# Adaptive dropout for training deep neural networks

**Lei Jimmy Ba    Brendan Frey**
Department of Electrical and Computer Engineering
University of Toronto
`jimmy, frey@psi.utoronto.ca`

## Abstract

Recently, it was shown that deep neural networks can perform very well if the activities of hidden units are regularized during learning, e.g, by randomly dropping out 50% of their activities. We describe a method called 'standout' in which a binary belief network is overlaid on a neural network and is used to regularize of its hidden units by selectively setting activities to zero. This 'adaptive dropout network' can be trained jointly with the neural network by approximately computing local expectations of binary dropout variables, computing derivatives using back-propagation, and using stochastic gradient descent. Interestingly, experiments show that the learnt dropout network parameters recapitulate the neural network parameters, suggesting that a good dropout network regularizes activities according to magnitude. When evaluated on the MNIST and NORB datasets, we found that our method achieves lower classification error rates than other feature learning methods, including standard dropout, denoising auto-encoders, and restricted Boltzmann machines. For example, our method achieves 0.80% and 5.8% errors on the MNIST and NORB test sets, which is better than state-of-the-art results obtained using feature learning methods, including those that use convolutional architectures.

## 1  Introduction

For decades, deep networks with broad hidden layers and full connectivity could not be trained to produce useful results, because of overfitting, slow convergence and other issues. One approach that has proven to be successful for unsupervised learning of both probabilistic generative models and auto-encoders is to train a deep network layer by layer in a greedy fashion [7]. Each layer of connections is learnt using contrastive divergence in a restricted Boltzmann machine (RBM) [6] or backpropagation through a one-layer auto-encoder [1], and then the hidden activities are used to train the next layer. When the parameters of a deep network are initialized in this way, further fine tuning can be used to improve the model, *e.g.*, for classification [2]. The unsupervised, pre-training stage is a crucial component for achieving competitive overall performance on classification tasks, e.g., Coates *et al.* [4] have achieved improved classification rates by using different unsupervised learning algorithms.

Recently, a technique called dropout was shown to significantly improve the performance of deep neural networks on various tasks [8], including vision problems [10]. Dropout randomly sets hidden unit activities to zero with a probability of $0.5$ during training. Each training example can thus be viewed as providing gradients for a different, randomly sampled architecture, so that the final neural network efficiently represents a huge ensemble of neural networks, with good generalization capability. Experimental results on several tasks show that dropout frequently and significantly improves the classification performance of deep architectures. Injecting noise for the purpose of regularization has been studied previously, but in the context of adding noise to the inputs [3],[21] and to network components [16].

Unfortunately, when dropout is used to discriminatively train a deep fully connected neural network on input with high variation, e.g., in viewpoint and angle, little benefit is achieved (section 5.5), unless spatial structure is built in.

In this paper, we describe a generalization of dropout, where the dropout probability for each hidden variable is computed using a binary belief network that shares parameters with the deep network. Our method works well both for unsupervised and supervised learning of deep networks. We present results on the MNIST and NORB datasets showing that our 'standout' technique can learn better feature detectors for handwritten digit and object recognition tasks. Interestingly, we also find that our method enables the successful training of deep auto-encoders from scratch, *i.e.*, without layer-by-layer pre-training.

## 2 The model

The original dropout technique [8] uses a constant probability for omitting a unit, so a natural question we considered is whether it may help to let this probability be different for different hidden units. In particular, there may be hidden units that can individually make confident predictions for the presence or absence of an important feature or combination of features. Dropout will ignore this confidence and drop the unit out 50% of the time. Viewed another way, suppose after dropout is applied, it is found that several hidden units are highly correlated in the pre-dropout activities. They could be combined into a single hidden unit with a lower dropout probability, freeing up hidden units for other purposes.

We denote the activity of unit $j$ in a deep neural network by $a_j$ and assume that its inputs are $\{a_i : i < j\}$. In dropout, $a_j$ is randomly set to zero with probability 0.5. Let $m_j$ be a binary variable that is used to mask, the activity $a_j$, so that its value is

$$a_j = m_j g\big(\sum_{i:i<j} w_{j,i} a_i\big), \tag{1}$$

where $w_{j,i}$ is the weight from unit $i$ to unit $j$ and $g(\cdot)$ is the activation function and $a_0 = 1$ accounts for biases. Whereas in standard dropout, $m_j$ is Bernoulli with probability 0.5, here we use an adaptive dropout probability that depends on input activities:

$$P(m_j = 1|\{a_i : i < j\}) = f\big(\sum_{i:i<j} \pi_{j,i} a_i\big), \tag{2}$$

where $\pi_{j,i}$ is the weight from unit $i$ to unit $j$ in the standout network or the adaptive dropout network; $f(\cdot)$ is a sigmoidal function, $f : \mathbb{R} \to [0,1]$. We use the logistic function, $f(z) = 1/(1 + \exp(-z))$.

The standout network is an adpative dropout network that can be viewed as a binary belief network that overlays the neural network and stochastically adapts its architecture, depending on the input. Unlike a traditional belief network, the distribution over the output variable is not obtained by marginalizing over the hidden mask variables. Instead, the distribution over the hidden mask variables should be viewed as specifying a Bayesian posterior distribution over models. Traditional Bayesian inference generates a posterior distribution that does not depend on the input at test time, whereas the posterior distribution described here does depend on the test input. At first, this may seem inappropriate. However, if we could exactly compute the Bayesian posterior distribution over neural networks (parameters and architectures), we would find strong correlations between components, such as the connectivity and weight magnitudes in one layer and the connectivity and weight magnitudes in the next layer. The standout network described above can be viewed as approximately taking into account these dependencies through the use of a parametric family of distributions.

The standout method described here can be simplified to obtain other dropout techniques. The original dropout method is obtained by clamping $\pi_{j,i} = 0$ for $0 \le i < j$. Another interesting setting is obtained by clamping $\pi_{j,i} = 0$ for $1 \le i < j$, but learning the input-independent dropout parameter $\pi_{j,0}$ for each unit $a_j$.

As in standard dropout, to process an input at test time, the stochastic feedforward process is replaced by taking the expectation of equation 1:

$$\mathbb{E}[a_j] = f\big(\sum_{i:i<j} \pi_{j,i} a_i\big) g\big(\sum_{i:i<j} w_{j,i} a_i\big). \tag{3}$$

We found that this method provides very similar results as randomly simulating the stochastic process and computing the expected output of the neural network.

## 3 Learning

For a specific configuration $m$ of the mask variables, let $L(m, w)$ denote the likelihood of a training set or a minibatch, where $w$ is the set of neural network parameters. It may include a prior as well.

The dependence of $L$ on the input and output have been suppressed for notational simplicity. Given the current dropout parameters, $\pi$, the standout network acts like a binary belief network that generates a distribution over the mask variables for the training set or minibatch, denoted $P(m|\pi, w)$. Again, we have suppressed the dependence on the input to the neural network. As described above, this distribution should not be viewed as the distribution over hidden variables in a latent variable model, but as an approximation to a Bayesian posterior distribution over model architectures.

The goal is to adjust $\pi$ and $w$ to make $P(m|\pi, w)$ close to the true posterior over architectures as given by $L(m, w)$, while also adjusting $L(m, w)$ so as maximize the data likelihood w.r.t. $w$. Since both the approximate posterior $P(m|\pi, w)$ and the likelihood $L(m, w)$ depend on the neural network parameters, we use a crude approximation that we found works well in practice. If the approximate posterior were as close as possible to the true posterior, then the derivative of the free energy $F(P, L)$ w.r.t $P$ would be zero and we can ignore terms of the form $\partial P/\partial w$. So, we adjust the neural network parameters using the approximate derivative,

$$-\sum_m P(m|\pi, w)\frac{\partial}{\partial w}\log L(m, w), \tag{4}$$

which can be computed by sampling from $P(m|\pi, w)$.

For a given setting of the neural network parameters, the standout network can in principal be adjusted to be closer to the Bayesian posterior by following the derivative of the free energy $F(P, L)$ w.r.t. $\pi$. This is difficult in practice, so we use an approximation where we assume the approximate posterior is correct and sample a configuration of $m$ from it. Then, for each hidden unit, we consider $m_j = 0$ and $m_j = 1$ and determine the partial contribution to the free energy. The standout network parameters are adjusted for that hidden unit so as to decrease the partial contribution to the free energy. Namely, the standout network updates are obtained by sampling the mask variables using the current standout network, performing forward propagation in the neural network, and computing the data likelihood. The mask variables are sequentially perturbed by combining the standout network probability for the mask variable with the data likelihood under the neural network, using a partial forward propagation. The resulting mask variables are used as complete data for updating the standout network.

The above learning technique is approximate, but works well in practice and achieves models that outperform standard dropout and other feature learning techniques, as described below.

**Algorithm 1:** Standout learning algorithm: alg1 and alg2

---

Notation: $H(\cdot)$ is Heaviside step function ;
**Input**: $w, \pi, \alpha, \beta$
alg1: initialize $w, \pi$ randomly; alg2: initialize $w$ randomly, set $\pi = w$;
**while** *not stopping criteria* **do**
    **for** *hidden unit* $j = 1, 2, ...$ **do**
        $P(m_j = 1|\{a_i : i < j\}) = f(\alpha \sum_{i:i<j} \pi_{j,i} a_i + \beta)$;
        $m_j \sim P(m_j = 1|\{a_i : i < j\})$;
        $a_j = m_j g(\sum_{i:i<j} w_{j,i} a_i)$;
    **end**
    Update neural network parameter $w$ using $\frac{\partial}{\partial w}\log L(m, w)$;
    `/* alg1`                                                              `*/`
    **for** *hidden unit* $j = 1, 2, ...$ **do**
        $t_j = H(L(m, w|m_j = 1) - L(m, w|m_j = 0))$
    **end**
    Update standout network $\pi$ using target $t$ ;
    `/* alg2`                                                              `*/`
    Update standout network $\pi$ using $\pi \leftarrow w$ ;
**end**

---

## 3.1 Stochastic adaptive mixtures of local experts

A neural network of N hidden units can be viewed as $2^N$ possible models given the standout mask $M$. Each of the $2^N$ models acts like a separate "expert" network that performs well for a subset of the input space. Training all $2^N$ models separately can easily over-fit to the data, but weight sharing among the models can prevent over-fitting. Therefore, the standout network, much like a gating network, also produces a distributed representation to stochastically choses which expert to

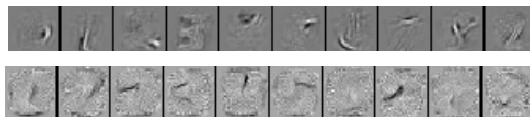

Figure 1: Weights from hidden units that are least likely to be dropped out, for examples from each of the 10 classes, for (top) auto-encoder and (bottom) discriminative neural networks trained using standout.

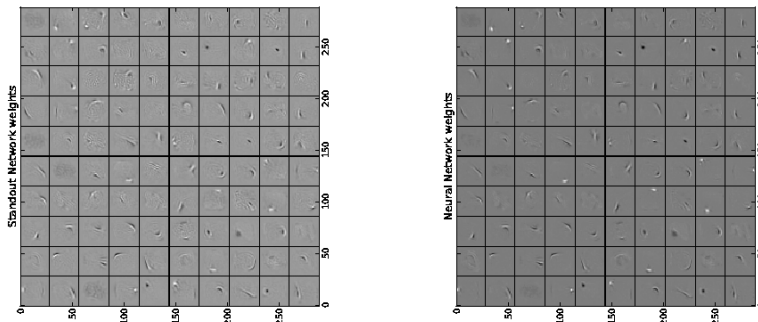

Figure 2: First layer standout network filters and neural network filters learnt from MNIST data using our method.

turn on for a given input. This means $2^N$ models are chosen by $N$ binary numbers in this distributed representation.

The standout network partitions the input space into different regions that are suitable for each expert. We can visualize the effect of the standout network by showing the units that output high standout probability for one class but not others. The standout network learns that some hidden units are important for one class and tend to keep those. These hidden units are then more likely to be dropped out when the input comes from a different class.

## 4 Exploratory experiments

Here, we study different aspects of our method using MNIST digits (see below for more details). We trained a shallow one hidden layer auto-encoder on MNIST using the approximate learning algorithm. We can visualize the effect of the standout network by showing the units that output low dropout probability for one class but not others. The standout network learns that some hidden units are important for one class and tends to keep those. These hidden units are more likely to be dropped when the input comes from a different class (see figure 1).

The first layer filters of both the standout network and the neural network are shown in figure 2. We noticed that the weights in the two networks are very similar. Since the learning algorithm for adjusting the dropout parameters is computationally burdensome (see above), we considered tying the parameters $w$ and $\pi$. To account for different scales and shifts, we set $\pi = \alpha w + \beta$, where $\alpha$ and $\beta$ are learnt.

Concretely, we found empirically that the standout network parameters trained in this way are quite similar (although not identical) to the neural network parameters, up to an affine transformation. This motivated our second algorithm *alg2* in psuedocode(1), where the neural network parameters are trained as described in learning section 3, but the standout parameters are set to an affine transformation of the neural network parameters with hyper-parameters alpha and beta. These hyper-parameters are determined as explained below. We found that this technique works very well in practice, for the MNIST and NORB datasets (see below). For example, for unsupervised learning on MNIST using the architecture described below, we obtained 153 errors for tied parameters and 158 errors for separately learnt parameters. This tied parameter learning algorithm is used for the experiments in the rest of the paper. In the above description of our method, we mentioned two hyper-parameters that need to be considered: the scale parameter $\alpha$ and the bias parameter $\beta$. Here we explore the choice of these parameters, by presenting some experimental results obtained by training a dropout model as described below using MNIST handwritten digit images.

$\alpha$ controls the sensitivity of the dropout function to the weighted sum of inputs that is used to determine the hidden activity. In particular, $\alpha$ scales the weighted sum of the activities from the

layer before. In contrast, the bias $\beta$ shifts the dropout probability to be high or low and ultimately controls the sparsity of the hidden unit activities. A model with a more negative $\beta$ will have most of its hidden activities concentrated near zero.

Figure 3(a) illustrates how choices of $\alpha$ and $\beta$ change the dependence of the dropout probability on the input. It shows a histogram of hidden unit activities after training networks with different $\alpha$'s and $\beta$'s on MNIST images.

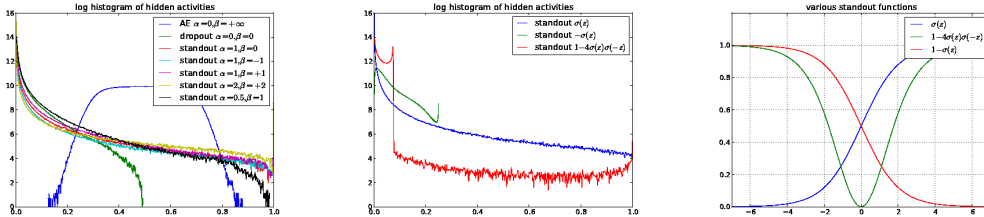

Figure 3: Histogram of hidden unit activities for various choices of hyper-parameters using the logistic dropout function, including those configurations that are equivalent to dropout and no dropout-based regularization (AE). Histograms of hidden unit activities for various dropout functions. Various standout function $f(\cdot)$

We also consider different forms of the dropout function other than the logistic function, as shown in figure 3(b). The effect of different functional forms can be observed in the histogram of the activities after training on the MNIST images. The logistic dropout function creates a sparse distribution of activation values, whereas the functions such as $f(z) = 1 - 4(1 - \sigma(z))\sigma(z)$ produce a multi-modal distribution over the activation values.

## 5 Experimental results

We consider both unsupervised learning and discriminative learning tasks, and compare results obtained using standout to those obtained using restricted Boltzmann machines (RBMs) and auto-encoders trained using dropout, for unsupervised feature learning tasks. We also investigate classification performance by applying standout during discriminative training using the MNIST and NORB [11] datasets.

In our experiments, we have made a few engineering choices that are consistent with previous publications in the area, so that our results are comparable to the literature. We used ReLU units, a linear momentum schedule, and an exponentially decaying learning rate (c.f. Nair *et al.* 2009[13]; Hinton *et al.* 2012 [8]). In addition, we used cross-validation to search over the learning rate (0.0001, 0.0003, 0.001, 0.003, 0.01, 0.03) and the values of alpha and beta (-2, -1.5, -1, -.5, 0, .5, 1, 1.5, 2) and for the NORB dataset, the number of hidden units (1000, 2000, 4000, 6000).

### 5.1 Datasets

The MNIST handwritten digit dataset is generally considered as a well-studied problem, which offers the ability to ensure that new algorithms produce sensible results when compared to the many other techniques that have been benchmarked. It consists of ten classes of handwritten digits, ranging from 0 to 9. There are, in total, 60,000 training images and 10,000 test images. Each image is $28 \times 28$ pixels in size. Following the common convention, we randomly separate the original training set into 50,000 training cases and 10,000 cases used for validating the choice of hyper-parameters. We concatenate all the pixels in an image in a raster scan fashion to create a 784-dimensional vector. The task is to predict the 10 class labels from the 784-dimensional input vector.

The small NORB normalized-uniform dataset contains 24,300 training examples and 24,300 test examples. It consists of 50 different objects from five different classes: cars, trucks, planes, animals, and humans. Each data point is represented by a stereo image pair of size $96 \times 96$ pixels. The training and test set used different object instances and images are created under different lighting conditions, elevations and azimuths. In order to perform well in NORB, it demands learning algorithms to learn features that can generalize to test set and be able to handle large input dimension. This makes NORB significantly more challenging than the MNIST dataset. The objects in the NORB dataset are 3D under difference out-of-plane rotation, and so on. Therefore, the models trained on NORB have to learn and store implicit representations of 3D structure, lighting and so on. We formulate

the data vector following Snoek *et al.*[17] by down-sampling from $96 \times 96$ to $32 \times 32$, so that the final training data vector has 2048 dimensions. Data points are subtracted by the mean and divided by the standard deviation along each input dimension across the whole training set to normalize the contrast. The goal is to predict the five class labels for the previously unseen 24,300 test examples. The training set is separated into 20,000 for training and 4,300 for validation.

## 5.2 Nonlinearity for feedforward network

We used the ReLU [13] activation function for all of the results reported here, both on unsupervised and discriminative tasks. The ReLU function can be written as $g(x) = \max(0, x)$. We found that its use significantly speeds up training by up to 10-fold, compared to the commonly used logistic activation function. The speed-up we observed can be explained in two ways. First, computations are saved when using max instead of the exponential function. Second, ReLUs do not suffer from the vanishing gradient problem that logistic functions have for very large inputs.

## 5.3 Momentum

We optimized the model parameters using stochastic gradient descent with the Nesterov momentum technique [19], which can effectively speed up learning when applied to large models compared to standard momentum. When using Nesterov momentum, the cost function $J$ and derivatives $\frac{\partial J}{\partial \theta}$ are evaluated at $\theta + v^k$, where $v^k = \gamma v^{k-1} + \eta \frac{\partial J}{\partial \theta}$ is the velocity and $\theta$ is the model parameter. $\gamma < 1$ is the momentum coefficient and $\eta$ is the learning rate. Nesterov momentum takes into account the velocity in parameter space when computing updates. Therefore, it further reduces oscillations compared to standard momentum.

We schedule the momentum coefficient $\gamma$ to further speed up the learning process. $\gamma$ starts at $0.5$ in the first epoch and linearly increase to $0.99$. The momentum stays at $0.99$ during the major portion of learning and then is linearly ramped down to $0.5$ during the end of learning.

## 5.4 Computation time

We used the publicly available *gnumpy* library [20] to implement our models. The models mentioned in this work are trained on a single Nvidia GTX 580 GPU. As in psuedocode(1), the first algorithm is relatively slow, since the number of computations is $O(n^2)$ where n is the number of hidden units. The second algorithm is much faster and takes $O(kn)$ time, where k is the number of configurations of the hyper-parameters alpha and beta that are searched over. In particular, for a 784-1000-784 auto-encoder model with mini-batches of size 100 and 50,000 training cases on a GTX 580 GPU, learning takes 1.66 seconds per epoch for standard dropout and 1.73 seconds for our second algorithm.

The computational cost of the improved representations produced by our algorithm is that a hyper-parameter search is needed. We note that some other recently developed dropout-related methods, such as maxout, also involve an additional computational factor.

## 5.5 Unsupervised feature learning

Having good features is crucial for obtaining competitive performance in classification and other high level tasks. Learning algorithms that can take advantage of unlabeled data are appealing due to increasing amount of unlabeled data. Furthermore, on more challenging datasets, such as NORB, a fully connected discriminative neural network trained from scratch tends to perform poorly, even with the help of dropout. (We trained a two hidden layer neural network on NORB to obtain 13% error rate and saw no improvement by using dropout). Such disappointing performance motived us to investigate unsupervised feature learning and pre-training strategies with our new method. Below, we show that our method can extract useful features in a self-taught fashion. The features extracted using our method not only outperform other common feature learning methods, but our method is also quite computationally efficient compared to techniques like sparse coding.

We use the following procedures for feature learning. We first extract the features using one of the unsupervised learning algorithms in figure (4). The usefulness of the extracted features are then evaluated by training a linear classifier to predict the object class from the extracted features. This process is similar to that employed in other feature learning research [14].

We trained a number of architectures on MNIST, including standard auto-encoders, dropout auto-encoders and standout auto-encoders. As described previously, we compute the expected value of

| | arch. | act. func. | err. |
|---|---|---|---|
| raw pixel | 784 | | 7.2% |
| RBM | 784-1000 weight decay | $\sigma(\cdot)$ | 1.81% |
| DAE | 784-1000-784 | $ReLU(\cdot)$ | 1.95% |
| dropout AE | 784-1000-784 50% hidden dropout | $ReLU(\cdot)$ | 1.70% |
| **standout AE** | 784-1000-784 standout | $ReLU(\cdot)$ | **1.53%** |

(a) MNIST

| | arch. | act. func. | err. |
|---|---|---|---|
| raw pixel | 8976 | | 23.6% |
| RBM | 2048-4000 weight decay | $\sigma(\cdot)$ | 10.6% |
| DAE | 2048-4000-2048 | $ReLU(\cdot)$ | 9.5% |
| dropout AE | 2048-4000-2048 50% hidden dropout | $ReLU(\cdot)$ | 10.1% |
| dropout AE * | 2048-4000-2048 22% hidden dropout | $ReLU(\cdot)$ | 8.9% |
| **standout AE** | 2048-4000-2048 standout | $ReLU(\cdot)$ | **7.3%** |

(b) NORB

Figure 4: Performance of unsupervised feature learning methods. The dropout probability in the DAE * was optimized using [18]

each hidden activity and use that as the feature when training a classifier. We also examined RBM's, where we the soft probability for each hidden unit as a feature. Different classifiers can be used and give similar performance; we used a linear SVM because it is fast and straightforward to apply. However, on a subset of problems we tried logistic classifiers and they achieved indistinguishable classification rates.

Results for the different architectures and learning methods are compared in table 4(a). The auto-encoder trained using our proposed technique with $\alpha = 1$ and $\beta = 0$ performed the best on MNIST.

We performed extensive experiments on the NORB dataset with larger models. The hyper-parameters used for the best result are $\alpha = 1$ and $\beta = 1$. Overall, we observed similar trends to the ones we observed for MNIST. Our standout method consistently performs better than other methods, as shown in table 4(b).

## 5.6 Discussion

The proposed standout method was able to outperform other feature learning methods in both datasets with a noticeable margin. The stochasticity introduced by the standout network successfully removes hidden units that are unnecessary for good performance and that hinder performance. By inspecting the weights from auto-encoders regularized by dropout and standout, we find that the standout auto-encoder weights are sharper than those learnt using dropout, which may be consistent with the improved performance on classification tasks.

The effect of the number of hidden units was studied using networks with sizes 500, 1000, 1500, and up to 4500. Figure 5 shows that all algorithms generally perform better by increasing the number of hidden units. One notable trend for dropout regularization is that it achieves significantly better performance with large numbers of hidden units since all units have equal chance to be omitted. In comparison, standout can achieve similar performance with only half as many hidden units, because highly useful hidden units will be kept more often while only the less effective units will be dropped.

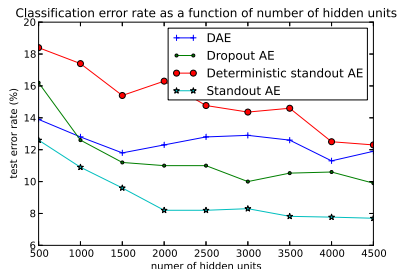

Figure 5: Classification error rate as a function of number of hidden units on NORB.

One question is whether it is the stochasticity of the standout network that helps, or just a different nonlinearity obtained by the expected activity in equation 3. To address this, we trained a deterministic auto-encoder with hidden activation functions given by equation 3. The result of this 'deterministic standout method' is shown in figure 5 and it performs quite poorly.

It is believed that sparse features can help improve the performance of linear classifiers. We found that auto-encoders trained using ReLU units and standout produce sparse features. We wondered whether training a sparse auto-encoder with a sparsity level matching the one obtained by our method would yield similar performance. We applied an L1 penalty on the hidden units and trained an auto-encoder to match the sparsity obtained by our method (figure4). The final features extracted using the sparse auto-encoder achieved 10.2% error on NORB, which is significantly worse than our method. Further gains can be achieved by tuning hyper-parameters, but the hyper-parameters for our method are easier to tune and, as shown above, have little effect on the final performance. Moreover, the sparse features learnt using standout are also computationally efficient compared

| | error rate |
|---|---|
| RBM + FT | 1.24% |
| DAE + FT | 1.3% |
| shallow dropout AE + FT | 1.10% |
| deep dropout AE + FT | 0.89% |
| **standout shallow AE + FT** | **1.06%** |
| **standout deep AE + FT** | **0.80%** |

(a) MNIST fine-tuned

| | error rate |
|---|---|
| DBN [15] | 8.3% |
| DBM [15] | 7.2% |
| third order RBM [12] | 6.5% |
| dropout shallow AE + FT | 7.5% |
| dropout deep AE + FT | 7.0% |
| **standout shallow AE + FT** | **6.2%** |
| **standout deep AE + FT** | **5.8%** |

(b) NORB fine-tuned

Figure 6: Performance of fine-tuned classifiers, where FT is fine-tuning

to more sophisticated encoding algorithms, e.g., [5]. To find the code for data points with more than 4000 dimensions and 4000 dictionary elements, the sparse coding algorithm quickly becomes impractical.

Surprisingly, a shallow network with standout regularization (table 4(b)) outperforms some of the much larger and deeper networks shown. Some of those deeper models have three or four times more parameters than the shallow network we trained here. This particular result show that a simpler model trained using our regularization technique can achieve higher performance compared to other, more complicated methods.

## 5.7 Discriminative learning

In deep learning, a common practice is to use the encoder weights learnt by an unsupervised learning method to initialize the early layers of a multilayer discriminative model. The backpropagation algorithm is then used to learn the weights for the last hidden layer and also fine tune the weights in the layers before. This procedure is often referred to as discriminative fine tuning. We initialized neural networks using the models described above. The regularization method that we used for unsupervised learning (RBM, dropout, standout) is also used for corresponding discriminative fine tuning. For example, if a neural network is initialized using an auto-encoder trained with standout, the neural network will also be fine tuned using standout for all its hidden units, with the same standout function and hyper-parameters as the auto-encoder.

During discriminative fine tuning, we hold the weights fixed for all layers except the last one for the first 10 epochs, and then the weights are updated jointly after that. As found by previous authors, we find that classification performance is usually improved by the use of discriminative fine tuning.

Impressively, we found that a two-hidden-layer neural network with 1000 ReLU units in its first and second hidden layers trained with standout is able to achieve 80 errors on MNIST data after fine tuning (error rate of 0.80%). This performance is better than the current best non-convolutional result [8] and the training procedure is simpler. On NORB dataset, we similarly achieved 6.2% error rate by fine tuning the simple shallow auto-encoder from table(4(b)). Furthermore, a two-hidden-layer neural network with 4000 ReLU units in both hidden layers that is pre-trained using standout achieved 5.8% error rate after fine tuning. It is worth mentioning that a small weight decay of 0.0005 is applied to this network during fine-tuning to further prevent overfitting. It outperforms other models that do not exploit spatial structure. As far as we know, this result is better than any previously published results without distortion or jitter. It even outperforms carefully designed convolutional neural networks found in [9].

Figure 6 reports the classification accuracy obtained by different models, including state-of-the-art deep networks.

## 6 Conclusions

Our results demonstrate that the proposed use of standout networks can significantly improve performance of feature-learning methods. Further, our results provide additional support for the 'regularization by noise' hypothesis that has been used to regularize other deep architectures, including RBMs and denoising auto-encoders, and in dropout.

An obvious missing piece in this research is a good theoretical understanding of why the standout network provides better regularization compared to the fixed dropout probability of 0.5. While we have motivated our approach as one of approximating the Bayesian posterior, further theoretical justifications are needed.

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
