[Reviews · NeurIPS 2013]

Submitted by Assigned_Reviewer_5

The paper proposes a novel dropout ("standout") approach to regularization in deep networks. In contrast to the familiar dropout (recently popularized by successes of convolutional deep networks in work by Hinton and colleagues) the probability of a unit to drop out is estimated by a parametric model, using the network weights. This is shown to be equivalent to a belief network overlaying the main feed-forward network. The paper describes a method for jointly learning the parameters of the main network and this additional dropout belief network, and reports on empirical evaluation showing that this is potentially a more efficient regularization technique.

I liked the paper quite a bit. It is well written, and the central idea seems clever, original, and timely. It makes a lot of sense that a data-driven dropout would be more efficient.
Of course, much of the proposed framework is ad-hoc (as admitted on ll. 424-427) but I don't hold it against the paper, since this is currently the nature of much of the cutting edge research on deep networks. Given the huge surge of interest in those, I am sure this paper will generate significant interest.

I have a couple of minor comments:

- It is unfortunate (but, again, common to the field) that the experiments involve a fairly large number of engineering decisions (number of units, values of learning rates, tweaking of schedule of updating some weights in some epochs or not) that would be hard to translate to different experiments/applications. So in that sense the field remains largely a form of art/craft.

- The experiments are sufficient to arouse interest in the proposed ideas, although hardly significant (MNIST digits seem to have outlived their usefulness since many methods now achieve statistically indistinguishable error rates near zero, and NORB is somewhat artificial in nature). It's OK by me since advancing state of the art on any particular application is not the point here.

- There seems to be a connection to mixture of experts architecture (hinted at on l. 98) and if so it would be good to discuss it in a bit more depth.

- A cartoon level figure illustrating different architectures used in experiments would be very helpful.

Summary: Nice paper, on a fundamental question (regularization) in a highly active subfield (deep learning), introducing novel and potentially important idea.

Submitted by Assigned_Reviewer_6

Summary:

"Adaptive dropout for training deep neural networks" describes and evaluates
a new neural network architecture that combines a sigmoid belief network and
a conventional feedforward network.

Pro:
* very strong results on MNIST and NORB
* novel model that generalizes dropout to full neural networks

Con:
* little theoretical insight
* empirical results somewhat scattered
* training algorithm not perfectly clear


Quality:

The strongest element of this work, to me, is the empirical results. The
performance of this non-convolutional model on MNIST and NORB is impressive.

The intuition that the stochastic units implement some Bayesian sampling over
models doesn't inspire me personally, but I don't object to the hypothesizing.

The learning algorithm appears to require more computation per iteration than
standard backprop and SGD, the authors should add some discussion of training
speed to the results / discussion.

The experimental results are not obviously focussed on any particular claim.
For example, the proposed method is a stochastic neural network, but it is
never simply trained as a supervised model. Why is it trained first as an
autoencoder? How much autoencoder training was necessary? Several
hyperparamters governing the optimization process appear to have been tuned,
were these tuned separately for the different model architectures? If they
weren't, then what can the reader draw from the model comparison scores? For
me, the experiment section mainly says that using your model, Nesterov
momentum, AE pre-training, and careful hyperparameter optimization, you got
a tweaked our fully connected model to get .8 on MNIST and 5.8 on NORB.

Using \alpha and \beta to tie \pi to 'w' and then also using ReLU units for
the non-stochastic parts of the model makes it feel like there's a much
simpler way to explain what you've done: namely, you have used a new
activaction function that is a ReLU with some stochastic noise added around
the inflection point. This may well be a good way (like maxout?) to get the
learning benefits of linear units without losing so many units because they
just turn "off" and can't come back on again.


Clarity:

The paper is well-organized, following the standard presentation for papers of
this type.

"principal" -> principle
"each experts" -> each expert

Figures 1 and 2 are not clear to me, what am I supposed to see? Are the
images in Figure 1 sorted by class label for example? Why does the leftmost
one look nothing like a 0 or a 1, and the rightmost one look nothing like
a 9 or 0.

In describing the learning algorithm, could the authors explain the procedure
for training \pi in more detail? It isn't clear to me. What is the free energy here,
and how is it possible to estimate each m_j's contribution fast enough for learning?

Section 4 is confusing, because after sketching one procedure for training \pi
in Section 3, this Section throws it away and makes \pi an affine
transformation of 'w'. It's not very good to say "we found this technique
works very well in practice", I'd like to see the evidence for myself (that's
what this paper's for). If it doesn't make any difference in terms of
end-of-training accuracy, then why did you tie them?

Why are not showing your CIFAR-10 results?

In line 190, you say that \alpha and \beta are "learnt" but then in line 194 these
are described as hyper-parameters. Which is it?


If \pi and w are tied by \alpha and \beta, then wouldn't a single "activation
function" plot be a clearer way to explain the effect of \alpha and \beta than
Figure 3? Figure 3 is tough to interpret for me, I don't know what to make of
activation histograms.


Originality:

The proposed model is novel.


Significance:

At least some members of the NIPS community love new stochastic/neural
architectures that work well and no one really understands why, that faction
will be intrigued by this paper.
Summary: Paper presents a new model that generalizes dropout training to multilayer networks, and improves best known non-convolutional scores on MNIST and NORB, but learning algorithm is not explained especially clearly and there is very little theoretical insight.

Submitted by Assigned_Reviewer_7

Summary.

The paper introduces a new method called standout. The idea: a binary belief network is overlaid on a neural network, and is used to decrease the information content of its hidden units by selectively setting activities to zero. It's basically a more controlled way of applying dropout to a neural network (Krizhevsky et al., 2012), by learning the dropout probability function for each neuron.

Quality / Clarity.

The work and report are of good quality, and the description of the model is clear.

Originality / Significance.

The dropout paper from Krizhevsky et al., 2012, has had quite an impact on the community (very general idea, which I believe has already helped lots of people improve their results / models). I expect many papers to build on this idea, and this is of course of them. Where most derivative papers would focus on understanding what dropout does, this paper focuses on extending the core idea by making the probability function learned, and dependent on the input activations of each neuron. Being myself quite interested and intrigued by dropout, I think this is a paper worth publishing.
Summary: The paper extends the dropout algorithm by proposing to learn the dropout probability function. It is an elegant idea, with satisfying experimental results (performance improvement on two standard datasets, MNIST and NORB).
Author Feedback

Author rebuttal: We thank the reviewers for carefully reading our paper and providing helpful feedback. We are glad that in general the reviews are quite positive and that the reviewers appreciated our idea and the record-breaking empirical results. The quality scores are 8, 6 and 7 and the main feedback is that we should more clearly explain the algorithm and the details of the experiments, such as engineering decisions. As outlined below, we have modified the main text to address this feedback as well as the more minor concerns; we hope this information will assist the reviewers and area chair in making a decision.

The learning algorithms

We have modified the main text to more clearly explain how the algorithms work. In the first case, the standout network and the neural network have separate parameters, whereas in the second case the parameters are tied. A sample for the mask variables is drawn using the standout network, and then forward and backward propagations are used to compute the updates for connections to units that were not dropped out. For untied parameters, the standout network updates are obtained by sampling the mask variables using the current standout network, performing forward propagation in the neural network, and computing the data likelihood. The mask variables are sequentially perturbed by combining the standout network probability for the mask variable with the data likelihood under the neural network, using a partial forward propagation. The resulting mask variables are used as complete data for updating the standout network. We found empirically that the standout network parameters trained in this way are quite similar (although not identical) to the neural network parameters, up to an affine transformation. This motivated the second algorithm, where the neural network parameters are trained as described above, but the standout parameters are set to an affine transformation with hyperparameters alpha and beta. These hyperparameters are determined as explained below.


Speed of the algorithm

We have modified the main text to more clearly explain how the computation times compare to those of standard dropout. The first algorithm takes O(n^2) time, whereas the second algorithm takes O(kn) time where k is the number of hyper parameter settings and n is the number of hidden units. For an AE-784-1000-784 architecture, one epoch of tied-weight standout learning for 50,000 training cases takes 1.73 seconds on a GTX 580 GPU, in contrast to 1.66 seconds for standard dropout.



Experimental details and engineering choices


We have modified the main text to more clearly explain the engineering choices and hyperparameter search ranges, and also to clear up the confusion in section 4 about how alpha and beta were set. We made a small number of engineering choices that are consistent with previous publications in the area, so that our results are comparable to the literature. We used ReLU units, a two-layer architecture, a linear momentum schedule, and an exponentially decaying learning rate (c.f. Nair et al 2009; Vincent et al. 2010; Rifai et al 2011; Hinton et al 2012). We used cross-validation to search over hyperparameters, which included the number of hidden units (500,1000,1500,2000), the learning rate (0.0001, 0.0003, 0.001, 0.003, 0.01, 0.03) and the values of alpha and beta (-2, -1.5, -1, -.5, 0, .5, 1, 1.5, 2). Using cross-validation to select hyperparameters has been shown to translate across application domains, including vision, speech and natural language processing. Because we used fully-connected architectures, we didn't have to make choices for filter widths (Lee et al 2009) or recurrent connectivity (Socher et al 2011).



Pretraining vs discriminative training


In the initial manuscript, we only reported results obtained by standout using pretraining. We did perform experiments using discriminative training, but the classification performance of standout was not distinguishable from dropout. Here are the results:

Error rate on MNIST

model regularization Error
784-1000-1000-10 dropout 1.14 +/- 0.11
784-1000-1000-10 standout 1.17 +/- 0.07

Error rate on NORB

model regularization Error
8976-4000-4000-5 dropout 14.14 +/- 0.45
8976-4000-4000-5 standout 13.86 +/- 0.52


We note that without using a convolutional architecture, the discriminative training results for NORB are not very competitive, for both standout and dropout. We agree that it is important to report the above results and we have modified the main text accordingly.


Minor comments


Regarding the CIFAR-10 dataset, the reason that we didn't include it in this paper is that in order to obtain competitive results, a convolutional or patch-pooling architecture with special topology is needed, which is beyond the scope of the present paper. Nonetheless, for your interest, we have recently obtained an efficient GPU implementation of a convolutional standout architecture and we are able to achieve an error rate of 14.3% using two convolutional layers followed by logistic regression.


Regarding the comment on the relationship of standout to a mixture of experts, we will comment on this in the revised manuscript.


Reviewer_7: Your selection of "Low impact" seems to be inconsistent with the text of of your review: "It is an elegant idea, with satisfying experimental results (performance improvement on two standard datasets, MNIST and NORB)."


To address the request that we show plots of the activation and standout functions so as to understand the effect of the hyperparameters, we have included thumbnail plots in figure 3 in the revised manuscript.